# Ecological Drivers of Community Cohesion

Chaitanya S. Gokhale,[a] Mariana Velasque,[b,c] Jai A. Denton[b,d]

aResearch Group for Theoretical Models of Eco-evolutionary Dynamics, Department of Evolutionary Theory, Max Planck Institute for Evolutionary Biology, Plön, Germany
bGenomics and Regulatory Systems Unit, Okinawa Institute of Science and Technology, Onna-son, Japan
cExperimental Evolutionary Biology Lab, Monash University, Clayton, Australia
dWorld Mosquito Program, Institute of Vector-borne Disease, Monash University, Clayton, Australia

Chaitanya S. Gokhale and Jai A. Denton contributed equally to this work. Author order was determined by descending knowledge of the Marvel Cinematic Universe.

**ABSTRACT** From protocellular to societal, networks of living systems are complex and multiscale. Discerning the factors that facilitate assembly of these intricate interdependencies using pairwise interactions can be nearly impossible. To facilitate a greater understanding, we developed a mathematical and computational model based on a synthetic four-strain *Saccharomyces cerevisiae* interdependent system. Specifically, we aimed to provide a greater understanding of how ecological factors influence community dynamics. By leveraging transiently structured ecologies, we were able to drive community cohesion. We show how ecological interventions could reverse or slow the extinction rate of a cohesive community. An interconnected system first needs to persist long enough to be a subject of natural selection. Our emulation of Darwin's "warm little ponds" with an ecology governed by transient compartmentalization provided the necessary persistence. Our results reveal utility across scales of organization, stressing the importance of cyclic processes in major evolutionary transitions, engineering of synthetic microbial consortia, and conservation biology.

**IMPORTANCE** We are facing unprecedented disruption and collapse of ecosystems across the globe. To have any hope of mitigating this phenomenon, a much greater understanding of ecosystem dynamics is required. However, ecosystems are typically composed of highly dynamic networks of individual species. These interactions are further modulated by abiotic and biotic factors that vary temporally and spatially. Thus, ecological dynamics are obfuscated by this complexity. Here, we developed a theoretical model, informed by a synthetic experimental system, of Darwin's "warm little ponds." This cycling four-species system seeks to elucidate the ecological factors that drive or inhibit interaction. We show that these factors could provide an essential tool for avoiding the accelerating ecological collapse. Our study also provides a starting point to develop a more encompassing model to inform conservation efforts.

**KEYWORDS** mutualism, evolutionary dynamics, ecological processes, persistence, cycling, ecology, microbial systems, synthetic biology, theoretical biology

Complex living systems are composed of a web of interactions. These interactions are usually structured in space (i.e., alongside an environmental gradient or conditions) and time. Temporal and spatial environmental modifications can alter the direction and strength of interactions. For instance, tropical savannas are characterized by highly seasonal rainfall, with pronounced dry seasons of varying duration. The high irradiance and heat present during dry seasons can create a water deficit, causing dramatic changes in vegetation structure and associated biodiversity, resulting in widespread mortality (1, 2). Similarly, periods with high rainfall increase resource availability, vegetative growth, species diversity, and the interactive potential (3). Although rainfall has a strong and somewhat predictable effect on savannas' diversity and abundance,

Address correspondence to Chaitanya S. Gokhale, gokhale@evolbio.mpg.de, or Jai A. Denton, jai.denton@monash.edu.

The authors declare no conflict of interest.

this is not always the case, and it can even be associated with a decrease in species abundance (4). Here, we studied how communities of replicators, in the most general sense, from prebiotic molecules to species, persist in such dynamic ecological processes. We use the simplicity of minimal synthetic systems and corresponding theoretical models to achieve our aim.

The complexity of dynamic ecologies can significantly obstruct understanding of community properties such as temporal and spatial components, stability, and even the type of interaction that connects nodes (antagonistic, mutualistic, trophic, etc.). The loss of a single node can cascade through the network in unpredictable ways. One example is that of the "living dead" (5, 6). Although still alive in the environment, a node can lack the ecological circumstances that allow it to be reproductive. In some cases, such circumstances can be reversed with intervention (e.g., reforestation and construction of ecological corridors) (7); however, extinctions of mutualist partners typically lead to coextinction (mutualism coextinction) (8–10), as observed in trees that lost their pollinators (11). Further difficulties arise when interactions are dynamic, causing a significant indirect effect that can modify species diversity in a nonquantifiable manner, such as interference and facilitation (12–15). Such complex natural systems impede our understanding of critical environmental components, such as abundance, resilience, diversity, species interactions, and other ecological network properties.

To avoid issues associated with indirect, weak, and convoluted interactions across longer timescales or environments, it helps to partition species interactions into their ecological role using functional groups (16). Thus, node-rich networks can be expressed in terms of functional diversity rather than species. This approach aims to understand how functional groups influence diversity and how systems adapt to changes in conditions, including environmental degradation. Furthermore, functional networks can occupy multiple scales, from molecular interaction to planet-wide biomes. This generality allows exploration of how small changes can ripple throughout a network at one level, either biotic or abiotic.

Unpacking this complexity may seem like a Sisyphean task, but this challenge is what a growing body of interlinked, highly tractable theoretical and experimental models seeks to overcome (17–21). Despite being reductionist, these approaches elucidate core principles of complex community dynamics through tightly controlled interactions. Synthetic systems, and corresponding theoretical models, allow specific control of biotic and abiotic components, ecological structure, nutrients, and growth conditions that enable precise measurement of the interaction outcomes and dynamics (18). Several synthetic microbial systems and closely linked theoretical models have been developed in this case. They fall into two categories: those with fixed functional diversity (17, 22, 23) and those where the function of individual components is permitted to evolve (20, 21), becoming evolutionarily stable. As these systems continue to improve and expand, we envision systems of greater complexity combining both categories.

The vast complexity of interactions across scales of organization inspires the theory developed herein. We focused, however, on the ecological factors at the root of incipient communities. To facilitate this, we built on the above-mentioned frameworks by developing an experimentally informed theoretical model. We aimed to show that even if units of functional dependency exist, ecology plays a role in shaping community cohesion. This may include displacing biotic functional groups through fluctuating abiotic factors, facilitating novel interdependencies, or inducing changes in the relative levels of community members.

The degree of connectivity and the strengths of a microbial community can be called community cohesion (24). Such cohesion is crucial in understanding microbial population dynamics and is shaped by both abiotic and biotic factors. For example, community coalescence, where previously isolated communities come into contact with each other, has been shown to drive community cohesion (25–27). Although our system includes cycles of subdivision, isolation, and coalescence, we focused on the ecological process that drives such phenomena.

Specifically, we developed a theoretical model with four interacting components informed by our engineered four-strain cross-feeding interdependent yeast system.

The corresponding theoretical model is based on the classical hypercycle (28). The model of a hypercycle has been used to describe systems of reactants that catalyze the replication of each other with implications from the origins of life to ecosystems (29, 30). The experimental and theoretical systems posit that fluctuating ecological processes can assemble functionally dependent but unconnected components into stable communities. This process facilitates exploration of the complex dynamics between species' functional diversity and environmental context and how their combination shapes the interactions in the lineage that survives. Instead of evolving symbiotic interactions, we started with engineered interactions (17, 22). This approach allowed us to control one aspect of eco-evolutionary dynamics, the interactions, while exploring the ecology (31, 32). Using the growth properties of this system, we developed a theoretical model. We demonstrate how small ecological changes combined with community sampling and merging influence system survivorship. Moreover, we show how ecological interventions can stave off system-wide extinction.

## RESULTS

Evolutionary and ecological constraints drive symbiosis in nature (33). Hence, our model and subsequent analysis are divided to reflect the ecological and evolutionary dynamics underpinning functional interdependence. These two core processes were first studied independently and then combined to study the joint eco-evolutionary trajectory.

**Ecological processes. (i) Ecologies.** The strains $ADE\uparrow$, $TRP\uparrow$, $HIS\uparrow$, and $LYS\uparrow$ are defined by their ability to share one of four essential metabolites: adenine (ADE), tryptophan (TRP), histidine (HIS), or lysine (LYS). We also define the different possible ecologies by the presence or absence of these four essential metabolites. In each case, dark blue (ADE/$ADE\uparrow$), red (TRP/$TRP\uparrow$), light blue (HIS/$HIS\uparrow$), and yellow (LYS/$LYS\uparrow$) are used in the figures to signify either the metabolite-containing ecology or metabolite-producing strain. In addition, 16 distinct ecological conditions, from extremely poor (devoid of any metabolite) to rich (where all the metabolites are present) can be visualized (Fig. 1). We used mixtures of the aforementioned colors to denote mixtures in both strains and ecologies. To begin with, we assumed that all 16 ecologies have the same probability of being realized.

Resources can typically vary spatially as well as temporally. We included this variability in our model by considering different possible ecological conditions. For simplicity, we visualized this diversity in 96 cultures reflecting a 96-well-plate experimental approach (Fig. 1). We sampled the aforementioned 16 possible ecological states (here uniformly) to fill the 96 positions. We present a baseline comparison of our setup against a system with a single population (single well) in the supplemental material.

Next, we inoculated each ecology with a microbial community. Communities of strains were generated by sampling from a global pool of strains (Fig. 1). Because of their physical limitations, ecologies can support only a limited number of individuals. Hence, the carrying capacity of each of the wells on the plate was limited to $K$. The composition of each community could at most consist of four different strains. This combinatorics leads to 256 distinct possibilities from the pool. For inoculation, the community was scaled to $K/2$, i.e., half the carrying capacity. Thus, while group sizes (here proxied by the density $K/2$) were fixed, the compositions were heterogeneous.

**(ii) Within-well dynamics.** Each of the 96 wells was a combination of the ecological state and a community. If the metabolite requirements of all the strains in each community were satisfied, we categorized them as viable (Fig. 1). Within each well, the growth of each of the four strains is given by

$$\frac{dx_i}{dt} = x_i R_i(E, S) \left(1 - \frac{\sum_{k=1}^{4} x_k}{K}\right) \tag{1}$$

Here, $R_i(E,S)$ captures the different growth rates depending on the ecology of the well ($E$) and the strain composition of the community ($S$). Each metabolite could be either present or absent in each well. If present, the ecology could be the source of the

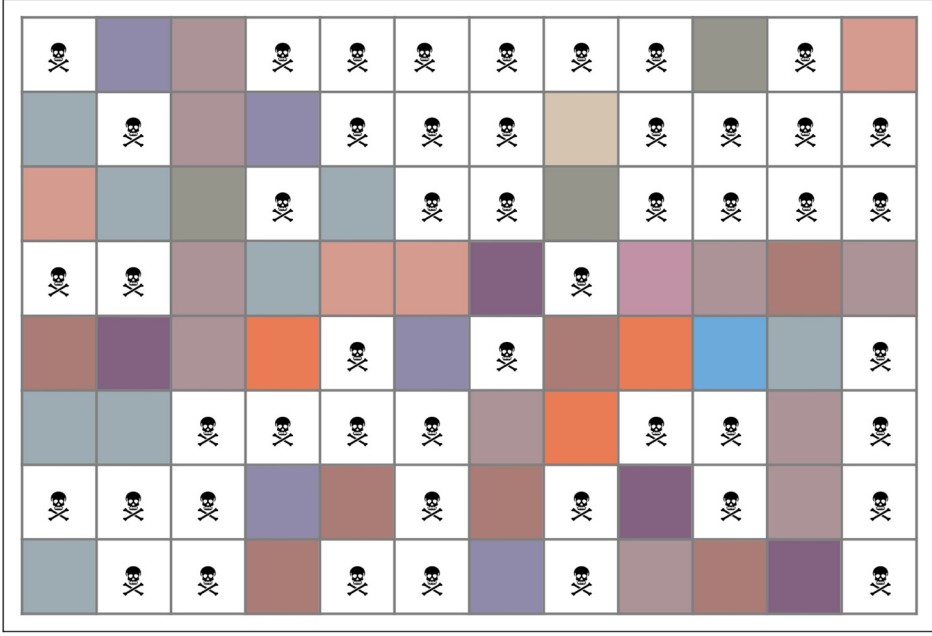

**FIG 1** Ecological niches and microbial community compositions. While the possible ecologies range from rich (1) to poor (0,0,0,0), the community compositions are a random sample of four types of strains. With the carrying capacity of each well designated K, we sampled a K/2 amount of

metabolite, a cross-feeding strain, or both. Growth rates of strain that rely on uptake, due to being unable to produce a specific nutrient, can be adversely impacted (34). For various such combinations of strains and ecologies, we inferred the relevant growth rates for each strain coming from experiments in Fig. S2 in the supplemental material.

All strains are assumed to have the same death rate ($d$). If a strain does not have access to all required metabolites, provided through either the community or ecology, it suffers this negative growth rate. In testing our model, we set $d$ at 0. This led to persistence, but this did not affect the qualitative result (https://github.com/tecoevo/ecoblocs). Thus, if the ecologies and communities do not match or if the strains themselves are mismatched, then those communities, as a whole, shrink (Fig. 1). The concentration phase in the wells lasts for $t_{wells}$ (time in wells) time steps.

**(iii) Cycling.** We implemented an ecological cycle to consider the dynamic nature of fluctuating environmental conditions. The cycle consisted of a concentration phase, represented by the 96 wells described above, and a pool phase, where the strains enjoyed a nutrient-rich environment unconstrained by space. At the end of the well phase, the concentrations were normalized, giving us the input for the next pool phase. In the pool, the strains grow exponentially,

$$\frac{dx_i}{dt} = r_i x_i \qquad (2)$$

where $r_i$ is the growth rate of strain $i$ in minimal medium, as given in Fig. S1. The duration of this process is designated $t_{pool}$. At the end of $t_{pool}$, we sampled the pool to prepare the inocula for the concentration phase. The ecologies and the communities for each of the 96 wells were prepared as per the procedure described above.

The result of an exemplary pool-well-pool cycle is shown in Fig. 2. As ecology and community sampling add stochasticity to the process, we show the average number of runs in the bottom panels of Fig. 2. The trajectories of the four strains show that the result overall is driven by pool fitnesses, where $HIS\uparrow$ has the highest fitness and thus dominates. Since there can be 16 different ecologies, the only ones able to support an all-$HIS\uparrow$ community are those in which (i) histidine is lacking and (ii) all metabolites are present, which can occur with a probability of 0.0625 each. Thus, the upper bound on the proportion of death that one can expect to see in repeated pool-well cycles over a long time is 0.875. Due to nonzero proportions of $ADE\uparrow$ and $LYS\uparrow$, the observed values are lower than expected.

**Ecological variations.** The time a community spends in a well is crucial in deciding if the population reaches appreciable densities by the end of the well stage. However, the carrying capacity of the well is a physical constraint that needs to be considered as a deciding factor in estimating the eventual frequencies of the strains transmitted to the pool into the next cycle. In the following sections, we discuss the effect of both the time spent in the well phase of the cycle and constrained carrying capacity.

**(i) Pool-well timescales.** The growth rates of the strains are a product of the community composition and the ecology. Thus, as described above, the strains have different growth rates in the pool and the wells. The amount of time the communities stay in the wells versus in pools drives the eventual strain abundances in the whole system. We observed that when the strains spent sufficient time in the pool phase ($t_{pool}$), the

**FIG 1** Legend (Continued)

strains. The resulting communities could be a mixture of all four strains ($K/8,K/8,K/8,K/8$) or a monoculture, e.g., only $ADE(\uparrow)$ ($K/2,0,0,0$), and all other possible combinations between. Only when the community members had access to all the required metabolites, either through cross-feeding or the ecology, was combination termed viable. For example, in the top left cell ({1,1}), the ecology is (1,1,0,0); i.e., only adenine and tryptophan are present in the ecology and the community consists of $ADE\uparrow$ and $TRP\uparrow$. The strains require histidine and lysine, which are not available, and hence the strains will die. In contrast, in cell {1,6}, the ecology is the same but the community composition consists of $ADE\uparrow$, $TRP\uparrow$, and $LYS\uparrow$, and this is thus is a viable cell. The difference in growth rate comes from the fact that cross-fed strains have a different growth rate than supplemented ones (e.g., see cells {6,7} and {6,8}). In each case, dark blue (ADE/$ADE\uparrow$), red (TRP/$TRP\uparrow$), light blue (HIS/$HIS\uparrow$), and yellow (LYS/$LYS\uparrow$) are used to signify either the metabolite-containing ecologies (top left) or metabolite-producing strains (top right). We used mixtures of these colors to denote mixtures in both strains and ecologies.

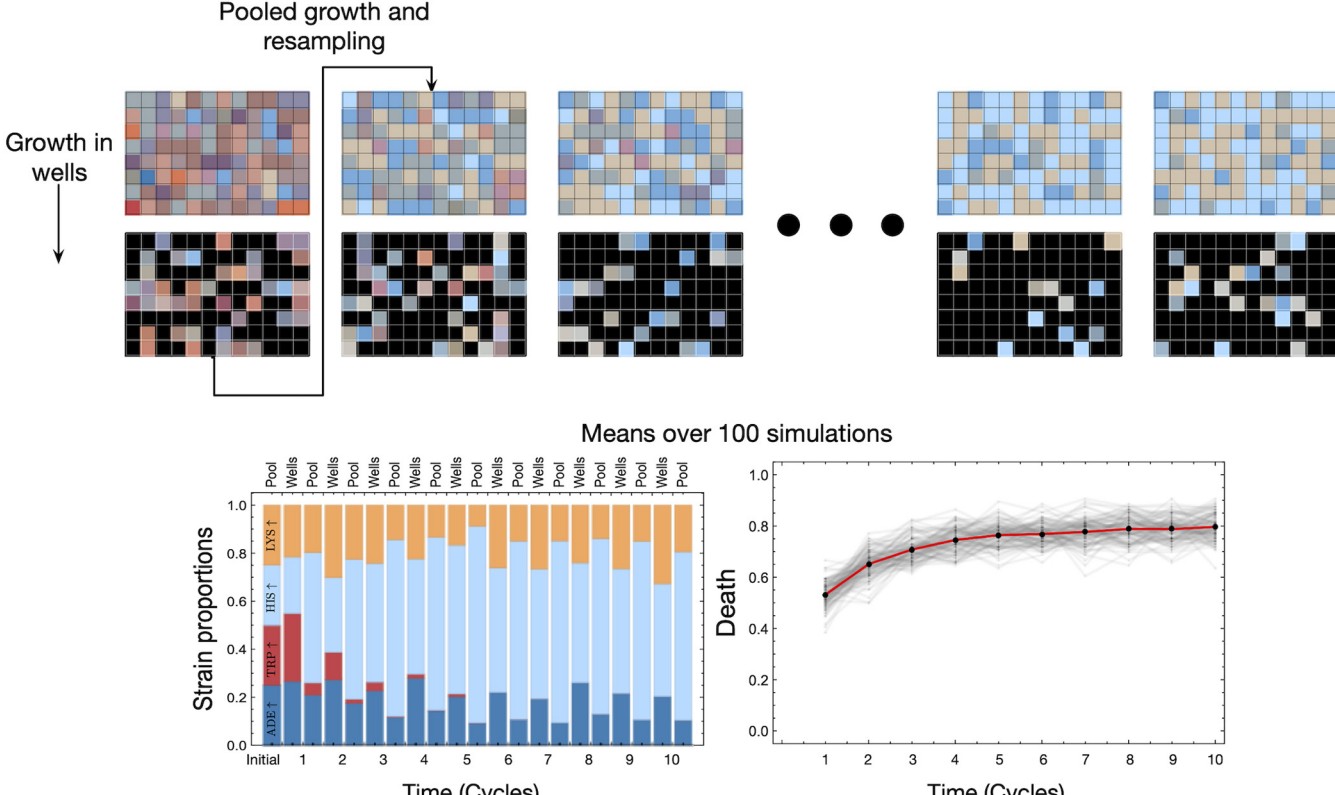

**FIG 2** Cycling between a pool phase and subsampled wells. As with tidal cycles in intertidal zones, the strains experience two environments periodically: the pool phase, which reflects the growth dynamics as governed by the rates given in Fig. S1, and the well phase. The wells have a carrying capacity ($K$) of 10 and are inoculated with small communities sampled at an amount equivalent to $K/2$. The strains grow with different growth rates in the wells (as a combination of the community structure and the well ecology), $R_i(E,S)$ (Fig. S2). If a community is not viable, then all strains in that community experience the same negative growth rate ($d = -2$). For the illustrative example above, the time the communities spend in the pool phase is short compared to the amount of time spent in the wells ($t_{pool} = 2$; $t_{wells} = 20$). As shown in the top panels, the strains cycle between the pool and the well phase. Over time, the global composition of the population changes (an example of a single run is shown in the bar chart, and a mean over 100 runs is given at bottom left), affecting the number of possible ecologies it can survive (given by the mean observed death rate in the 96 wells [bottom right]). All the runs start with all strains present in equal proportions (0.25).

effect of the fitnesses in the well phase of the cycle was negligible (Fig. S6). Even though the growth rates in the wells are a complex combination of the community composition and the ecology, if the time spent in the wells was sufficiently short, then only the pool growth rates determined the population equilibrium. Essentially, the effect of the structured ecology would then be negligible and inconsequential to the long-term dynamics of the system. The emergence of complex interactions between the constituent strains was then not observed, since the simple single-peaked fitness landscape was driving the dynamics (Fig. S6).

**(ii) Dependence on carrying capacity.** If the time spent in the well phase is not a constraint, then final strain proportions are dictated by the carrying capacity of the wells. Differences in strain densities due to various carrying capacities are exemplified in Fig. 3. Starting from the same initial strain proportions and densities, for a larger carrying capacity (Fig. 3, second column), we observed *HIS↑* taking over the well. Thus, when sampling was based on pool representation, *HIS↑* would be overrepresented. This overrepresentation of one strain increases the likelihood of extinction in the next generation due to a lack of diversity. Although the lack of produced metabolites can be compensated for by viable ecologies, the slight increase in wells that showed death captures the effect of the increased carrying capacity. When *HIS↑* takes over not only the well but eventually the pool, then the only environments that can support the strain are {1,1,0,1} and {1,1,1,1}. Since each environment has the same probability of being realized, as mentioned above, the probability that a randomly selected well will go extinct is 0.87. Even

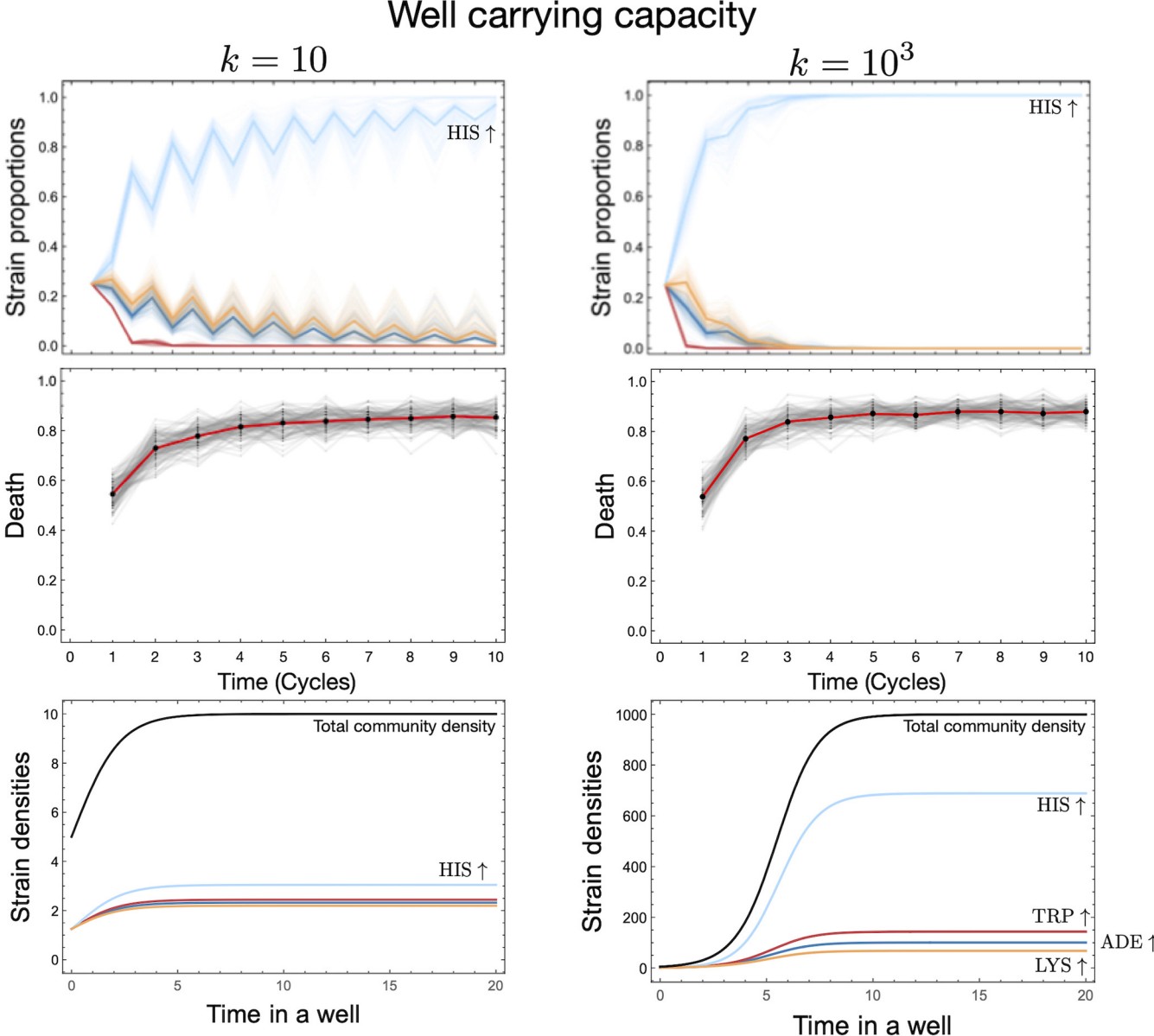

**FIG 3** Dependence on carrying capacity of a well. The carrying capacity of the well translates into the population size that a community can achieve during the time spent in the well. The result of two different carrying capacities can be seen in the difference in the time required for the *HIS↑* strain to take over the global population. This also elevates the death rate observed at the community level, as the dominance by one strain reduces resilience. The strain proportions and the mean death were averaged over 100 runs, where, for each run, all strains are present in equal proportions (0.25). To visualize the precise effect of the carrying capacity, we plotted example dynamics from a particular well (poor ecology with no metabolites) containing a community consisting of all strains in equal proportions. The growth rate for this setup extrapolated from the experiments (Fig. S1) is {0.804, 0.869, 1.157, 0.732} for *ADE↑*, *TRP↑*, *HIS↑*, and *LYS↑*.

for a finite sample of 96 wells, our wells approached this mean death rate when the environments were equally likely.

However, our assumption that nutrients are available uniformly over the ecological space and hence that all ecologies are equally probably is highly unrealistic. Furthermore, metabolites produced by each community can also feed into wells and the pool. This ecological feedback of information between cycles leads to our next section, thus completing the loop between the strains and their environment.

**Ecological feedback. (i) Niche availability.** There were 16 possible ecological conditions, given the four metabolites that can be mixed and matched in our system. For the wells, the ecologies were chosen randomly from these 16 possibilities in equal proportion. Given the realistic diversity of nutrients in space and time, we encapsulated

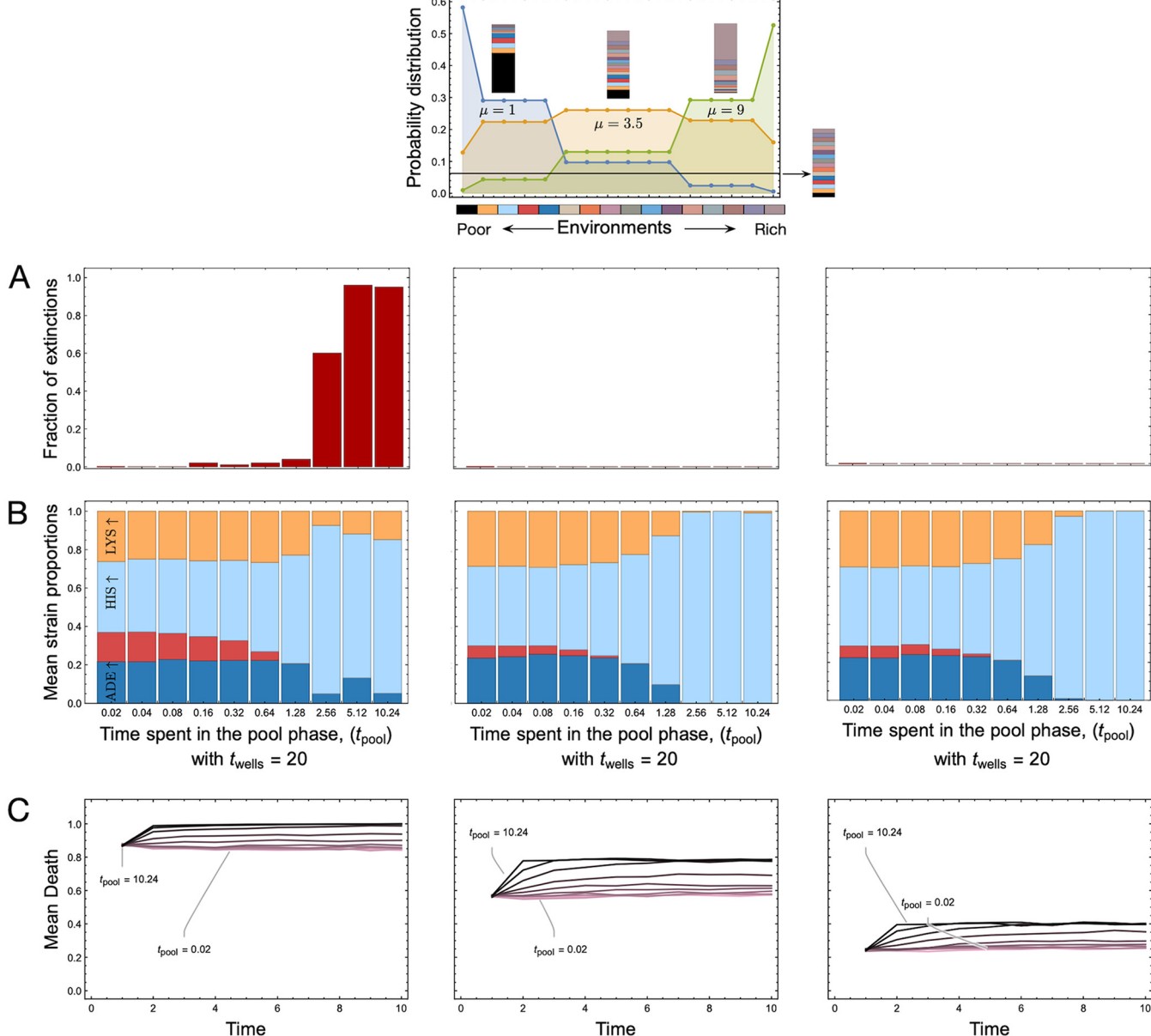

**FIG 4** Long-term dynamics for different ecological distributions. Until now, we had assumed that the 16 possible ecological variants seen in Fig. 1 were equally likely. Here, we relaxed this assumption and assessed the likelihood of observing a particular ecology (according to richness) with a probability drawn from a Poisson distribution with different means ($\mu = 1$, 3.5, and 9). (A) Final mean equilibrium frequencies (averaged over 100 runs) after 10 cycles of pool and well phases. The time spent in the wells ($t_{wells}$) is set to 20, and the time spent in the pool is varied. (B) Mean strain proportions across the 100 runs for different amounts of time spent in the pool phase. In all cases, the *HIS*↑ strain increases in proportion as $t_{pool}$ increases. (C) Mean observed death rates for the different values of $t_{pool}$, which reaches a maximum given by the inverse of the probability of observing the only two permissive environments for *HIS*↑, namely, {1,1,0,1} and {1,1,1,1}. Hence with increasing Poisson mean, as the distribution skews toward the right, we see an overrepresentation of the two environments (denoted by environments 14 and 16) allowing the survival of the system.

this stochasticity in the ecologies by changing the probability distribution used to draw the random ecologies. While, earlier, all 16 possibilities were equally likely, now we used a Poisson distribution over the discrete space of the number of ecologies with different means (Fig. 4). We drew from three ecological distributions of the 16 possible nutrient combinations skewed toward single or low numbers of nutrients, a medium number of nutrients, or heavily overrepresented high numbers of nutrients.

**(ii) Niche stabilization.** Organisms typically not only find a niche but also shape it. We have not yet considered the impact of the strains on the ecology itself. With each strain producing metabolites, the environment was altered in the wells (Fig. S4). The compositional

changes in the strain population thus directly affected the metabolite distributions. Realistically, the metabolite distributions were not purely exogenously determined but were also affected by the change due to the strains. Therefore, this effect needed to be captured explicitly via metabolic feedback. Biasing the distributions of metabolites according to the strains that survived in the previous cycle would ultimately affect the probabilities of observing the possible ecologies in the next cycle. For example, if the $ADE\uparrow$ strain was overrepresented in a community, there would be an overproduction of adenine. Thus, the choice of ecologies in the next selection round might be biased toward ecologies containing adenine. Hence, the changed metabolite concentrations distorted the distribution of ecologies, as seen in the next generation of the well phase. In the experiment described in the section above, we selected the ecology from a pre-set distribution. In the current case, we set the distribution itself as given by the distribution of the strains in the previous pool phase.

We take as an example the case where the difference in the timescales between the pool and well phases was of an order of magnitude ($t_{wells} = 20$ while $t_{pool} = 1.28$). In this case, the $HIS\uparrow$ strain is overrepresented, with almost 87% of the wells going extinct on average. We could allow metabolic feedback to begin at different time points in the season. We show in Fig. 5 that if we started the feedback later in the cycling, then the $HIS\uparrow$ strain had already reached appreciable frequencies. Therefore, in the next cycle, most of the ecologies in the wells would have an overabundance of histidine and minuscule amounts of other metabolites. This skew in ecologies hindered the growth of the most abundant strain, $HIS\uparrow$, itself and led to higher death rates in the whole system.

**(iii) Avoiding collapse.** Altering growth rates, regardless of the process, can promote community stability. This can include indirect and direct factors. By reducing $t_{pool}$, and thereby reducing the benefit of a higher growth rate, the system has a higher probability of surviving (Fig. 6, top left). Thus, any ecological factors that resulted in effective growth equalization reduced the likelihood of collapse. This includes the aforementioned $t_{pool}$ phase modification but also potentially any sustained changes to the ecological cycling. Alternatively, we show that direct equalization of strain growth rates also decreases extinctions (Fig. 6, bottom left). This direct effect can be caused by ecological factors altering growth rate, such as temperature, pH, or nutrients. It could also include evolutionary forces or direct genetic modification that reshape community growth rates. Regardless of the contributing processes of avoiding extinctions, almost all the lineages survived early interventions, and the interventions at intermediate or later stages were less lethal (Fig. 6). For synthetic systems, we envision that both of these processes can be engineered. Finally in both cases, as seen in Fig. 5, early intervention greatly improved survival (Fig. 6, right column).

## DISCUSSION

Cyclical variation in ecology affects interactions, population dynamics, abundance, and richness in a range of systems (35, 36). Despite this knowledge, the fundamental role of environmental periodicity is understudied (37). We conceived a stable evolutionary system with fixed functional diversity operating on short timescales to understand how cyclic ecological processes influence the abundance and community formation of organisms in a biological system. This system showed how cyclical fluctuations in resource availability and population-resource feedback loops shape network interactions. Our model embeds a four-strain interaction network in an environment cycling between structured and unstructured phases to explore how ecological processes shape community formation, cohesion, and persistence. The importance of the structured ecology is highlighted when contrasted with the null model (Text S1), as presented in Fig. S5, where extinction is guaranteed. We established how cyclic nutrient availability generates spatial and temporal heterogeneity and is capable of altering resource exploration, modifying selective pressure on ecologically relevant interactions (from cooperation to competition), and potentially serving as the basis for macroevolutionary dynamics.

Our system comprises cyclical compartmentalization of replicators and resources.

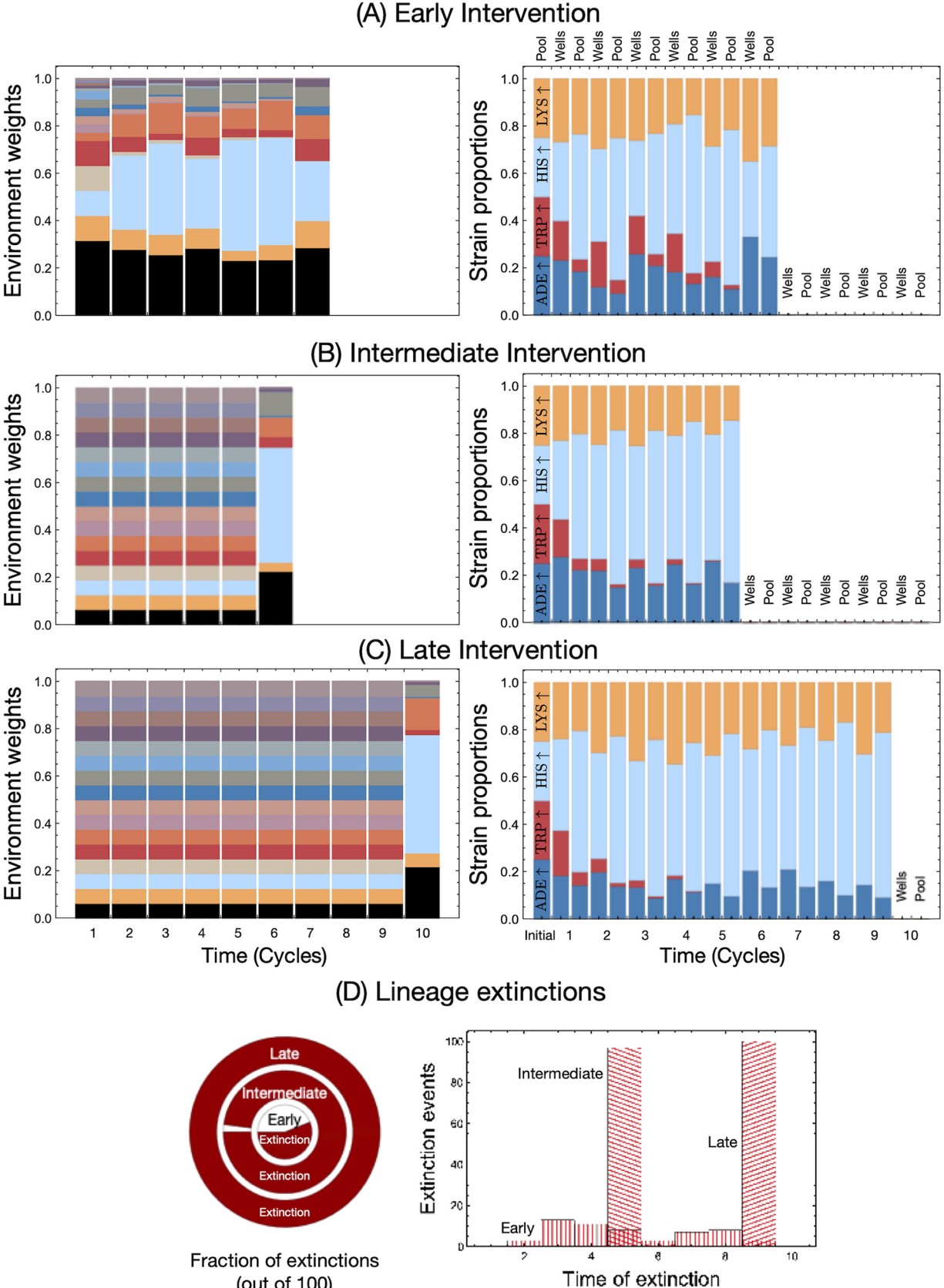

**FIG 5** Population dynamics under ecological feedback of metabolites. Until now we assumed that the ecological cycles exclude the metabolite transport from one cycle to the next. However, focusing only on interstrain dynamics provides only a partial picture. Here, we

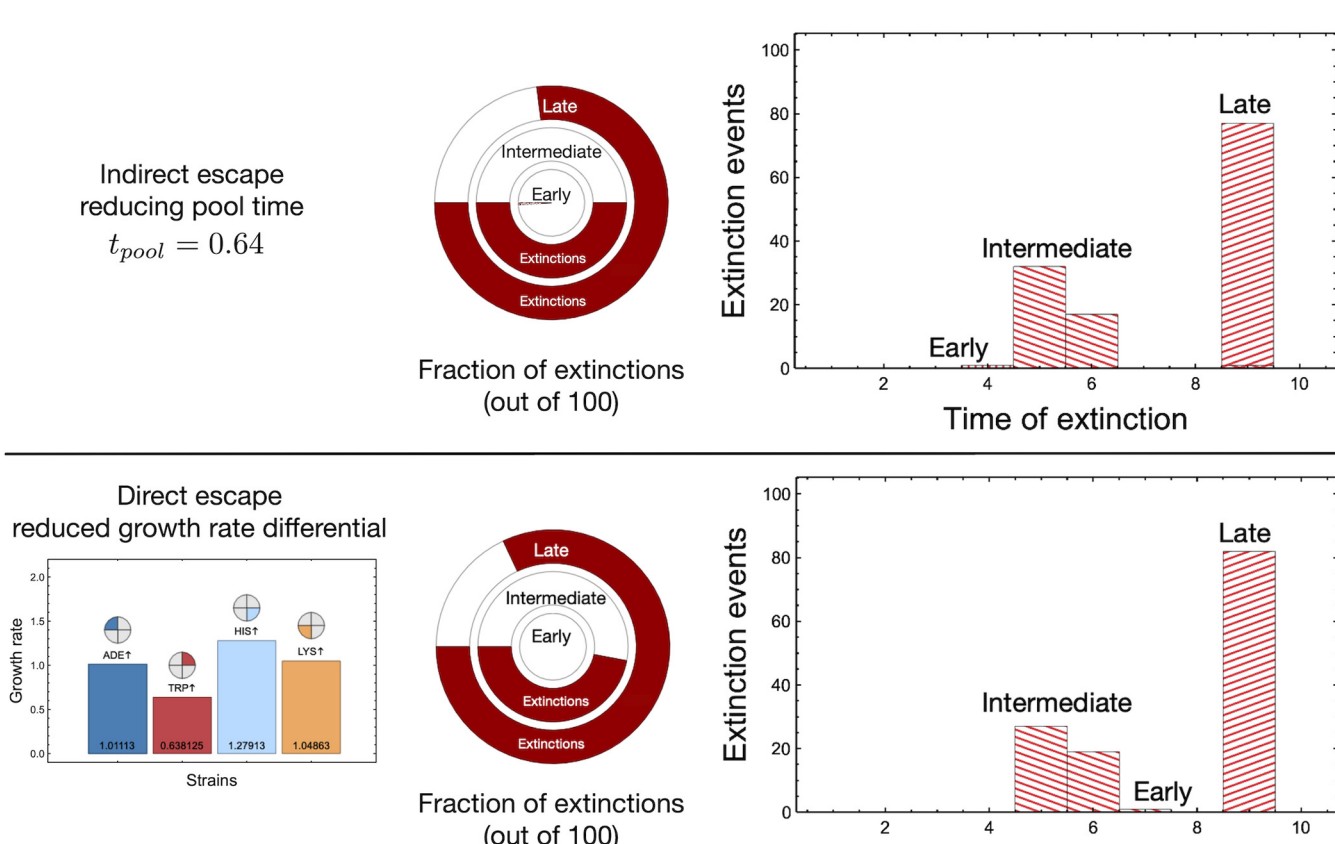

**FIG 6** Avoiding lineage collapse under ecological fluctuations. Equalization, either directly or indirectly, of growth rates allows avoidance of community collapse at the lineage level. From Fig. 5D, we observe that for intermediate and late interventions, all lineages go extinct. (Top) If the amount of time spent in the pool phase was shorter (half that shown in Fig. 5D), thereby reducing advantages afforded by higher growth rates, we rescued a substantial fraction of lineages in both the intermediate and late intervention treatments. (Bottom) Rescue is possible if the growth rates of the strains are quantitatively less different from each other. For example, here, we brought the growth rates closer to the means for all strains by 50% compared to their previously calculated rates. This change maintains the differences between the growth rates qualitatively, but a decrease in the number of extinctions is visible.

Essential resources, such as nutrients, water, and space, often fluctuate in regular cycles linked to major forces like the movement of the earth. Hence, living organisms are subject to changes in conditions that affect their survival and reproductive potential. The process is, however, more general, extending to prelife (38). At the level of prebiotic molecules, the structure of alkaline hydrothermal vents is pockmarked with the essential nutrients of autocatalytic reactions flowing in and out through them in a heterogeneous manner (39). Alternatively, for the RNA world hypothesis, transient compartmentalization and dry-wet cycles of the "warm little ponds" are postulated to be necessary for polymerization to generate complexity (38, 40). At the level of individual organisms, species adapted to live on rocky shores are subject to periodic changes in conditions caused by receding tides, which can lead to increased mortality and modify population structure and species abundance (41, 42). Although cyclical events usually occur alongside a con-

**FIG 5** Legend (Continued)

show the results of allowing for a connection between community population dynamics and ecology with respect to metabolite concentrations. From Fig. S6 we focus on a particular example where $t_{pool}$ is set to 1.28 and $t_{wells}$ is set to 20. The resulting pool frequencies drive the environmental sampling for the next well cycle. (A to C, left) Distributions from which the ecologies were chosen (at each well step); (right) resulting dynamics of single runs. If we allow the feedback process to start from the beginning, there is a higher possibility that the lineage survives (A) as opposed to starting the intervention later (C). (D) For the three interventions, the fractions of extinctions as calculated over multiple (56.98 and 100 of 100) runs are presented in the pie chart, and the time distribution of extinctions is in the graph.

tinuum, compartment-based models can provide a better alternative to its study. We used resource availability and defined carrying capacity to create distinct environmental compartments. Each cycle (compartmentalized and homogeneous) acted as an environmental sieve that led to changes in population density and frequency, which carried over to the next cycle (Fig. 2).

Our implementation of cyclical compartmentalization, the pool-well-like dynamics, is highly prevalent in naturally occurring systems subject to regular occurrences such as tides, seasons, or monsoons. In the compartmentalization of transient estuaries and intertidal zones, niche size and timing are widely known to alter eco-evolutionary dynamics. For example, trait evolution is linked to compartment size in transient estuaries (43). Furthermore, temporal compartmentalization has led to different ecotypes of the marine midge *Clunio* (44). However, in these real-world situations, niches within each compartment are not random. They are shaped by existing factors and thus are a subset of the broader systems. Our four-component experimental system exhibited substantial growth differences between engineered strains. Therefore, long growth periods in the nutrient-rich pool phase, without interdependencies, exacerbated these growth differences and reduced the representation of weaker strains. Thus, compartmentalization that limits such unrestricted growth reduces the impact of strain growth differences. However, increased well size can offset this dependence on compartmentalization (Fig. 3).

Our system, employing fixed interactions and nutrient limitations, revealed that establishment of a given species depends on the pool composition in previous cycles, indicating that even in simple systems, ecological dynamics are temporally and spatially interconnected. Furthermore, we also demonstrated the role of temporal heterogeneity in creating seasonal niches. Therefore, this can directly mediate functional diversity and influence the ecological dynamics between the interactive communities (45, 46). For instance, limited and cyclical resources maximized ecosystem stability by increasing the organism's functional role and interdependence between cooperating strains (Fig. 4). In comparison, constant high resource availability had the opposite effect. It eroded interdependence and promoted competition between strains (47, 48), which led to a population expansion, followed by the dominance of a single species. Once ecological conditions were no longer favorable, the dominant species showed a reduced viability, leading to an increased death rate (Fig. 4) and eventually extinction. This highlights that disruption of naturally occurring cyclical cycles can result in ecosystem collapse. This can be observed in aquatic dead zones, which are water bodies where life cannot survive due to low oxygen levels, with one of their most common causes being eutrophication. Excess phosphorus and nitrogen from fertilizer runoff or sewage cause an excessive algal bloom, depleting dissolved nutrients in the water and oxygen and leading to the death of other aquatic organisms and, eventually, the algae (49–51).

As niches with lower diversity have greater interstrain dependence, rapid extinction was observed with minimal scope for nutrients to compensate for the loss of even a single strain. Thus, with our system having fixed and nonoverlapping biological functions, each strain was critical to optimal community survival. That effectively made the weakest strain, *TRP*↑, a keystone species (the effect of removing all TRP-containing environments is illustrated in Fig. S7 and explained in Text S1). Essentially all aspects of a given biological system are impacted by a continuum of natural processes occurring at every level of the organization. Engineered interdependencies enforce this relationship and allow us to dissect biological concepts like keystone species. It was impossible to predict the ecological impact of reintroducing wolves to Yellowstone Park due to the enormous number of uncharacterized interactions and the subsequent effects of these interactions (52). However, with defined fixed interactions, we can model ecological outcomes in simplified systems, e.g., the explicit linking of the loss of weaker but critical species to ecological collapse.

Although factors that help preserve strain diversity stave off ecosystem collapse, we sought to explore further factors that prevented collapse. By allowing each pool-well cycle to influence the nutrient composition of the next cycle (Fig. 5), the strain-induced

environmental changes became a model for the Allee effect (53, 54). An early intervention, allowing intercycle feedback from the first cycle, maintained all four strains in most instances. The low frequency of any given strain reduced the corresponding nutrient availability and lowered overall population fitness (55) but provided an advantage to the underrepresented strain. In previous interactions in our model, complete extinction, even if only a single strain survived, was impossible due to the equal provision of nutrients. However, with intermediate and late intercycle interventions, complete extinction was immediate. Confirming the keystone species dynamics, the loss of any single strain always results in extinction with the intercycle interactions (Fig. S7). In addition to direct feedback, system collapse could be delayed by increasing the number of initial strains used to seed a given well. Moreover, migration into the system could potentially reduce the system's collapse and even resurrect dead lineages. However, neither process is expected to benefit the incorporated system feedback.

Manipulating ecological and evolutionary dynamics is a possible means of avoiding ecosystem-wide collapse (Fig. 6) (56–58). Within our system, any processes that would act to equalize growth rates would facilitate collapse avoidance. Ecologically, if the amount of time spent in the pool phase is reduced, the strain with the highest growth rate will not have enough time to outcompete others. Additionally, direct ecological factors constantly alter growth rates. For instance, growth rates can be altered by small daily or seasonal fluctuations in conditions, which are typically averaged or abstracted when modeling. However, extensive exploratory laboratory testing would further increase the granularity of growth estimations, determine additional factors improving community survival, and refine the model accordingly. However, reducing abstraction in this way would also reduce the broader applicability of the system. Differences in strain growth rates may also be reduced evolutionarily. Presently there is a strong tendency toward extinction primarily due to variable strain growth. This potentially causes strong selective pressure on evolutionary adaptation through equalization of growth rates and therefore allowing variability in growth rates within our system would be the first step toward explicit evolutionary processes. However, our simple experimental and theoretical model allows us to clarify and understand the effects of these two processes separately. This simplicity allows us to probe scenarios across complexity scales, from prebiotic chemistry at the inception of evolution to modern living systems where both processes act in tandem.

It can, however, be challenging to untangle causality in genetic and environmental factors (59). Thus, fixed biological functions are often used to remove the role genetics play (17, 22, 23). Incipient evolutionary processes must necessarily exclude the presence of interactions between replicators – from cells to species. One of the methods to impose selection is then ecological scaffolding, where the ecology selects on a specific property of the collective (60). In our study, we exemplified the role of ecology but did not assume any property of the community to select for but rather propagated what persisted (30). Persistence then provided a fertile test bed for evolutionary processes to develop, such as multilevel selection. Endogenization of cyclical ecological processes such as the development of life cycles and circadian and circalunar cycles may provide an evolutionary advantage to communities.

Although it was informed by a biological system, a fraction of growth parameter space was incorporated into our model. We focused on growth rates under nutrient-poor conditions that required rescue of auxotrophic mutations through medium supplementation or inclusion of an appropriate feedback-resistant (FBR) strain. However, complete nutritional rescues may be impossible for many auxotrophic mutations (61). This includes growth in rich media. Moreover, even well-studied auxotrophies can have cryptic pleiotropic phenotypes (62). However, this mutation class is a powerful tool facilitating the modeling of complex inter- and intraspecies dynamics. In addition, by limiting model growth parameters, we also removed the role of abiotic factors like temperature or pH in shaping ecology through their modulation of growth rates. Expansion of our framework to incorporate greater growth rate diversity would

provide greater accuracy. However, even in a laboratory-based system, such as yeast, obtaining an accurate representation of growth rates can prove difficult (63). In complex ecosystems or when greater parameter diversity is required, it can be impossible. Therefore, as with any model, a careful balancing act between abstraction and accuracy is needed.

Ecological networks can be highly variable concerning their constituents (64–66). However, our current model has only a limited capacity for modularization. Future work facilitating a greater understanding of complex eco-evolutionary network dynamics will require greater modularization by adding additional nodes and subnetworks. A logical first step is to develop experimental and theoretical systems that reflect greater strain diversity, specifically, strains that represent all 16 combinations of function, from sharing none of the metabolites to sharing all four. This system allows the formation of multiple possible consortia with the ability to survive. Additionally, this developed system will help test if greater diversity will allow for greater stability of lineages. We will thus be able to advance our understanding of nested ecological dynamics, consortium expansion, and cross-scale interaction.

This theoretical framework is a part of a larger ongoing project on the evolution and ecology of synthetic communities (22, 23, 32). An application of an extended consortium system, and a key driver behind the development of this model, would be predicting the outcomes of ecosystem interventions. For example, synthetic microbial communities with complex biological functions are being developed for fields in translational biology such as conservation, health, and exploration, from bioremediation and biodiversity restoration to space mining (67–70). We previously defined consortia with set inputs and outputs as swappable "ecoblocks" in synthetic microbiology (31). However, the role of ecology where such ecoblocks might be released is vital but so far neglected. Assessing the viability of such consortia using dynamic ecological principles, as was done here, or via in silico driven artificial selection will help design efficient synthetic consortia (71, 72). Once the communities have achieved their goal, self-limitation through community extinction might be a desired feature, not a bug (31).

## MATERIALS AND METHODS

**Experimental materials. (i) *S. cerevisiae* strains.** The synthetic system is composed of four metabolite-overproducing strains developed in the w303 strain background. A detailed description of strain construction is provided in the supplemental material. A strain list is available at https://github.com/tecoevo/ecoblocs/blob/main/strain_creation/strain_list.csv.

**(ii) Media.** Synthetic complete (SC) medium was made from dextrose, yeast nitrogen base, and Kaiser synthetic complete dropout supplement (Formedium, Norfolk, UK). Amino acid-free minimal medium (SC−aa) was made as described above without synthetic complete dropout supplement. Yeast extract-peptone-dextrose (YPD) medium was made using chemicals from BD Diagnostics (Tokyo, Japan) and Sigma (Kawasaki, Japan).

**Experimental procedures. (i) Culturing and sampling.** A comprehensive outline of the culturing methods employed is provided in the supplemental material. Very briefly, all cultures were grown in triplicate starting from a single colony at 30°C. They were grown to carrying capacity, washed, and left to grow without amino acid supplementation to induce starvation. The cells were then washed and diluted. Strains were mixed and supplemented as indicated. The cultures were sampled to determine growth and strain ratios every 24 h (including time zero) by removing 20 $\mu$L. Cell counts per milliliter of viable cells were obtained from experimental cultures using the Muse cell counter (Merck Millipore) with the Cell Count & Viability kit per the manufacturer's instructions. To obtain strain ratios, samples were serially diluted, plated on YPD solid medium, and allowed to grow for 2 days, and colonies were counted and replicated on the appropriate selective SC medium.

**(ii) Estimating growth parameters.** Growth constants underpin theoretical modeling but are highly complex in biological systems. Although increasingly complex derivatives can be employed to capture the temporal life history of a particular organism, these are approximations and increase the complexity of subsequent models. Moreover, it is impossible to generalize growth rates in response to biotic factors, such as other strains, and abiotic factors, such as nutrient availability, due to the enormous numbers of interdependencies that further complicate these complex temporal life histories. As such, while grounding our model in laboratory-based experiments, we have taken a high-level approach to the generation of a growth values that summarize these complex dynamics. Laboratory-based growth analysis was conducted for a large subset of growth conditions, with missing data estimated from these. Unfortunately, as we no longer have laboratory access, additional measurements have been impossible to obtain. A description of how these growth rates were calculated and an alternative method for their calculation are provided in the supplemental material (Text S1). The growth rates are shown in Fig. S2, and the

alternative method growth rates are included in Fig. S3. In addition, raw data and growth rates are provided on GitHub (https://github.com/tecoevo/ecoblocs).

**(iii) Strains.** Our system relies on four-way metabolite cross-feeding with one of the four strains, each containing an FBR mutation that causes overproduction of either adenine, tryptophan, histidine, or lysine. Several FBR mutations in numerous biosynthetic pathways have been identified that facilitate yeast cross-feeding (73). This has facilitated the development of synthetic yeast cross-feeding models of mutualism. A system of interdependent strains of yeast, one overproducing lysine and lacking adenine and the other overproducing adenine and lacking lysine, was developed in the study reported in reference 22 and analyzed in detail both biochemically and ecologically (17, 32). The strains overproducing leucine and lacking tryptophan and vice versa were developed in the study reported in reference 23.

Here, we developed a four-strain system that relied on the previously referenced FBR mutants. We constructed strains overproducing one of the metabolites while requiring the other three. For example, strain *ADE↑* contains a mutation in ADE4 that prevents the biosynthetic pathway from being inactivated. The resulting construct overproduces adenine and requires tryptophan, leucine, and histidine for growth. Similar mutations were employed in each of the four strains—*ADE↑*, *TRP↑*, *HIS↑*, and *LYS↑*—with all the metabolites present *ad libitum*, and the growth rates of these engineered strains were calculated (Fig. S1).

## SUPPLEMENTAL MATERIAL

Supplemental material is available online only.

**TEXT S1**, PDF file, 0.1 MB.
**FIG S1**, EPS file, 0.2 MB.
**FIG S2**, EPS file, 0.6 MB.
**FIG S3**, EPS file, 0.6 MB.
**FIG S4**, EPS file, 0.4 MB.
**FIG S5**, EPS file, 0.4 MB.
**FIG S6**, EPS file, 0.4 MB.
**FIG S7**, EPS file, 0.4 MB.

## ACKNOWLEDGMENTS

This work was supported by JSPS KAKENHI grant 19K06795 to J.A.D. and C.S.G. C.S.G. acknowledges support from the Max Planck Society. M.V. and J.A.D. were supported by the Okinawan Institute of Science & Technology and Monash University. The funders had no role in study design, data collection and interpretation, or the decision to submit the work for publication.

We thank the laboratories of D. Greig, W. Shou, and A. Murray for providing yeast strains.

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
