## [Reviewer comments · mSystems]

Ecological Drivers of Community Cohesion

Jai Denton, Chaitanya Gokhale, and Mariana Velasque

Corresponding Author(s): Jai Denton, Monash University

Review Timeline:

Submission Date:	September 27, 2022
Editorial Decision:	October 23, 2022
Revision Received:	December 7, 2022
Accepted:	December 16, 2022

Editor: Babak Momeni

Reviewer(s): The reviewers have opted to remain anonymous.

Transaction Report:

DOI: <https://doi.org/10.1128/msystems.00929-22>

October 23, 2022

Dr. Jai Denton
Monash University
World Mosquito Program
Clayton
Australia

Re: mSystems00929-22 (Ecological Drivers of Community Cohesion)

Dear Dr. Jai Denton:

Thank you for submitting your manuscript to mSystems. We have completed our review and I am pleased to inform you that, in principle, we expect to accept it for publication in mSystems. However, acceptance will not be final until you have adequately addressed the reviewer comments. The reviewers in particular had concerns about the clarity of presentation and accuracy of language that I encourage you to address in your revision. Additionally, one of the reviewers brought up the concern that the system is a bit artificial. In your revisions, I encourage you to expand the discussions on how your findings can be generalized and applied to other systems beyond the particular system you have investigated.

Preparing Revision Guidelines

Sincerely,

Babak Momeni

Editor, mSystems

Journals Department
American Society for Microbiology

Reviewer comments:

Reviewer #1 (Comments for the Author):

In the paper "Ecological Drivers of Community Cohesion," the authors pose a mathematical model, parameterized by experimental data from a synthetic system of 4 *Saccharomyces cerevisiae* with metabolic interdependencies. This model addresses the relevant question of collapse and resilience of communities under cycles of environmental fluctuation.

They found that ecology is the major driving force for community resilience regardless of the metabolic interplays present in the system. This means that by just adding some amino acids to the environment, the authors were able to rescue the community from collapse.

I think this is a relevant paper that addresses a very relevant question and having a synthetic system allow the authors simplify the complex dynamics of environmental fluctuations.

General comments.

Lines 246-254 It is not clear how the authors introduce metabolic feedback in the model. Maybe the authors could rewrite this part explaining the implementation of the metabolic feedback more clearly. On the other hand, have introduced a "migration regime," meaning, more strains of the correct kind will produce the same effect as the proposed feedback? Could this be affected by density?

Lines 259-263 While I agree with the evolutionary statement as equalizing the growth rates of the strains, I think this statement is incomplete. The authors have equalized the growth rates but increased the growth rates it may happen because of (particularly in this type of strain that lacks a metabolic pathway) the redundancy of function because the strain is again functional in any environment. Discussing why selection would equalize the strains' growth rate would be a good addition to the discussion.

Although I understand the constraints that have to measure the growth rate experimentally with and without amino acids, it is important to notice that in this particular system, the growth rate could be different in the rich environment than in poor environments, or with added amino acids. Discussing this could be a good addition to the discussion.

Minor comments

Figure 4. There are no panels. The figure captions refer to panels A, B, and C.

Even if it is repetitive, there should be labels for the strains in all figures. Figure 2,4,5,6.

Over the discussion, a couple of figures are not correctly referenced. Lines 309, 336.

Reviewer #2 (Comments for the Author):

In the manuscript entitled "Ecological Drivers of Community Cohesion", Gokhale and the coauthors built a mathematical model of cross-feeding four strain communities. This model contains a spatial structure consisting of 96 wells, and each has a different environment. The growth of species in a well depends on the environment and the member of the community, while species can exponentially grow in a pool phase between two well phases. Then, the durations of the well phase and the pool phase affect the relative abundance of each strain. In addition, the probability distribution of the environmental condition (whether it is static as in Fig. 4, or it dynamically changes due to the feedback as in Fig.5) affects the extinction time of the communities.

I noticed some points that are not clearly explained, that I cannot agree with, or that needs additional analysis. In addition, some figures do not seem to match the legend or the main text.

Major comments

1. Niche availability and Fig.4 As shown in Fig.1, the authors ordered 16 environmental conditions depending on the presence/absence of four metabolites. However, the orders of the environments are arbitrary, and using the Poisson distribution in Fig. 4 to generate the probability distribution is problematic. I agree that the environment (0,0,0,0) is the poorest and (1,1,1,1)

is the richest, but why do the authors assume (1,0,0,0) is richer than (0,1,0,0), (0,0,1,0) and (0,0,0,1)? Unless there is a reason that we can assume ADE increases the environmental richness more than TRP, HIS, and LYS, we can swap the indexes of the environments where the number of present metabolites is identical. If this comment is valid, the authors should generate the probability distribution of the environments in another way. For example, we can first determine the number of metabolites in a well (we could use Poisson distribution in this step). Next, we decide which metabolites exist in each well given a probability distribution that each of metabolite is chosen.

2. L. 246-254 and Fig. 5. It is unclear how the authors implemented the feedback between the communities and distributions of ecologies. They verbally explained it in l.239-245, but I do not think readers can understand how such feedback is implemented in the model. For example, the authors could explain how the probability distribution of the environments at cycle T is given by that distribution at cycle T-1 and the distributions of the communities. This point could affect the interpretation of why the timing of starting feedback between the environments and communities affects the extinction time.

3. I cannot understand why the authors suddenly talk about the evolution from l. 259. In the models, the authors define only ecological (population) dynamics, but not evolutionary dynamics. The legend of Fig. 6 explains the case where the four species have closer growth rates (or smaller fitness differences). However, such changes in the growth rates could occur other than evolution (e.g., change in growth temperature or pH), and there is no reason to assume that evolution can decrease the differences in growth rates. For these reasons, I do not think that Fig. 6 is interpreted as the evolutionary rescue.

4. I think the authors lack a negative control where there is no spatial structure (i.e., there exists only one well). If the environment is fixed as (1,1,1,1), I guess the sky blue strain (HIS) dominates and this strain would not go extinct. However, if the environment fluctuates with some probability as in Fig.4 by the feedback as in Fig.5, how long does the community survive (I mean, when all of the four species go extinct)? Such control simulations would deepen the understanding of the results of Figs. 4 -6 since the spatial structure affects the species coexistence and extinction.

Minor comments

1. It seems that the colors in Fig.1 does not match the explanations in the legend. For example, the authors wrote the ecology (or the environment) of the top left well (I believe {1,1}) is (1,0,0,0). According to the color bar on the top of Fig. 1, this should correspond to blue, but the color of the circle on the left column of Fig. 1 is purplish (I believe this environment is (1,1,0,0)). The colors at {1,4} do not match the legend either. I believe I am not color-blind. Please check the figure colors and the legend.
2. I would like the authors to clarify the definition of the strains (l. 127-128, l. 138-139, Fig. 1). Do the colors in the right column of Fig. 1 correspond to which metabolites are produced by each strain or community? Although the main text defines the strains by which metabolite each strain produces, Fig.1 does not explicitly explain what the colors of the communities (strains) mean.
3. The font sizes in the pie charts in Figs. 5 and 6 are too small to read. I am young (in academia), but it is very hard to read the letters there.
4. I do not think what is written in l. 256-258 matches the top panel of Fig. 6. Fig. 6 shows the effect of the duration of the pool phase, while the text explains the effect of the timing of the feedback.

Typos

- l. 122 "Nature" => "nature" (I do not think the authors are talking about the specific journal)
 - l. 123 "analysis, are" => "analysis are"
- The left-hand-side of Eq (1) should be dx_i/dt . Also, this equation lacks the death rate d .
The left-hand-side of Eq (2) should be dx_i/dt .

Response to Reviewers

NOTE: All line numbers refer to diff.pdf.

Reviewer #1:

General comments.

Lines 246-254 It is not clear how the authors introduce metabolic feedback in the model. Maybe the authors could rewrite this part explaining the implementation of the metabolic feedback more clearly.

A concern regarding the clarity of this section was also raised by Reviewer 2 (comment 2). We have rewritten this and added to the subsection "Niche stabilisation" where we introduce metabolic feedback concept. Also, we have added a figure in the supplementary materials that shows which parts of the model are affected by this metabolic feedback (figure SI.4) and included the code (mathematica and pdf format) on GitHub that facilitates feedback tuning. We hope this greatly improves the readability and reproducibility of this section.

On the other hand, have introduced a "migration regime," meaning, more strains of the correct kind will produce the same effect as the proposed feedback? Could this be affected by density?

This is an excellent point. We didn't explore the parameter space around well density and initial population size to a large degree nor did we explore migration at all. We feel that changes in density and initial strain seeding would only serve to delay a collapse but migration could provide a source to revive dead lineages. This would be quite an exciting addition to the model. We have mentioned this in our discussion (lines 403-407). One of our proposed projects moving forward includes both migration and evolution of novel strains.

Lines 259-263 While I agree with the evolutionary statement as equalizing the growth rates of the strains, I think this statement is incomplete. The authors have equalized the growth rates but increased the growth rates it may happen because of (particularly in this type of strain that lacks a metabolic pathway) the redundancy of function because the strain is again functional in any environment. Discussing why selection would equalize the strains' growth rate would be a good addition to the discussion.

Also raised by Reviewer 2 (comment 3), this section poorly articulated our key message. Our aim was to highlight that even under a fluctuating environment, system collapse can be avoided. One of the simplest ways is through changes in strain growth rates. Although we focused on evolutionary changes, any factors that alter growth would be equally as effective.

We updated this section to reflect this (lines 277-297). Moreover, with respect to your comment just below, we have added greater discussion of growth rates generally (lines 408-444).

Although I understand the constraints that have to measure the growth rate experimentally with and without amino acids, it is important to notice that in this particular system, the growth rate could be different in the rich environment than in poor environments, or with added amino acids. Discussing this could be a good addition to the discussion.

We have added an additional paragraph discussing limitations of any growth rate analysis. Essentially, we capture a snapshot of the enormous parameter space of a given strain's growth (lines 443-457). Also how the growth of a strain is measured itself can be a challenge (Challenges and pitfalls of inferring microbial growth rates from lab cultures Ana-Hermina Ghenu, Loïc Marrec, Claudia Bank, *bioRxiv* 2022.06.24.497412; doi: <https://doi.org/10.1101/2022.06.24.497412>). Given that we are deriving GxE fitnesses, the situation is further complicated. An alternative method of fitness calculation as derived in the SI (Fig SI 2 vs 3) was also tested but results in no difference in the conclusions of our work (full data and code available in the GitHub repository). We concur that further systematic expansion of the parameter space would be a potential inclusion in versions of our system.

Minor comments.

Figure 4. There are no panels. The figure captions refer to panels A, B, and C.

Thank you for catching this. We have fixed the figure.

Even if it is repetitive, there should be labels for the strains in all figures. Figure 2,4,5,6.

All the strains have been labelled in the figures.

Over the discussion, a couple of figures are not correctly referenced. Lines 309, 336.

Apologies! These were references to the supplementary materials that didn't compile. Both have been corrected.

Reviewer #2 (Comments for the Author):

In the manuscript entitled "Ecological Drivers of Community Cohesion", Gokhale and the coauthors built a mathematical model of cross-feeding four strain communities. This model contains a spatial structure consisting of 96 wells, and each has a different environment. The growth of species in a well depends on the environment and the member of the community, while species can exponentially grow in a pool phase between two well phases. Then, the durations of the well phase and the pool phase affect the relative abundance of each strain. In addition, the probability distribution of the environmental condition (whether it is static as in Fig. 4, or it dynamically changes due to the feedback as in Fig.5) affects the extinction time of the communities.

I noticed some points that are not clearly explained, that I cannot agree with, or that needs additional analysis. In addition, some figures do not seem to match the legend or the main text.

Major comments

1. Niche availability and Fig.4 As shown in Fig.1, the authors ordered 16 environmental conditions depending on the presence/absence of four metabolites. However, the orders of the environments are arbitrary, and using the Poisson distribution in Fig. 4 to generate the probability distribution is problematic. I agree that the environment (0,0,0,0) is the poorest and (1,1,1,1) is the richest, but why do the authors assume (1,0,0,0) is richer than (0,1,0,0), (0,0,1,0) and (0,0,0,1)? Unless there is a reason that we can assume ADE increases the environmental richness more than TRP, HIS, and LYS, we can swap the indexes of the environments where the number of present metabolites is identical. If this comment is valid, the authors should generate the probability distribution of the environments in another way. For example, we can first determine the number of metabolites in a well (we could use Poisson distribution in this step). Next, we decide which metabolites exist in each well given a probability distribution that each of metabolite is chosen.

Our treatment of environments with identical numbers of nutrient supplementations as varying in quality based on arbitrary nutrient order was undertaken for display purposes. It allowed us to highlight the influence environmental quality has on community health. However, upon reflection, we completely agree that is highly misleading without an extensive series of caveats. Therefore, we have implemented the changes you suggested (figure 4). This includes in the main figure and in the available code on GitHub. The change resulted in only slight changes to the wording and no changes to the interpretation of the result.

2. L. 246-254 and Fig. 5. It is unclear how the authors implemented the feedback between the communities and distributions of ecologies. They verbally explained it in l.239-245, but I do not think readers can understand how such feedback is implemented in the model. For example, the authors could explain how the probability distribution of the

environments at cycle T is given by that distribution at cycle T-1 and the distributions of the communities. This point could affect the interpretation of why the timing of starting feedback between the environments and communities affects the extinction time.

A concern regarding the clarity of this section was also raised by Reviewer 1. Given the importance of this concept in the interpretation of original figure 5 (now figure 6), we have endeavoured to greatly improve the clarity through the addition of more explicit exploratory text (lines 280-297) and a diagram outlining feedback in our model (figure SI.4).

3. I cannot understand why the authors suddenly talk about the evolution from l. 259. In the models, the authors define only ecological (population) dynamics, but not evolutionary dynamics. The legend of Fig. 6 explains the case where the four species have closer growth rates (or smaller fitness differences). However, such changes in the growth rates could occur other than evolution (e.g., change in growth temperature or ph), and there is no reason to assume that evolution can decrease the differences in growth rates. For these reasons, I do not think that Fig. 6 is interpreted as the evolutionary rescue.

Once again a similar concern was also raised by Reviewer 1. This highlights just how jarring the switch to *evolutionary rescue* was to close the results section. Our initial aim was to highlight that even under a fluctuating environment, system collapse can be avoided. One of the simplest ways is through changes in strain growth rates. However, as you mentioned this can be achieved through any number of non-evolutionary ways. We have articulated this in the text (lines 277-297) Moreover, we added greater discussion on factors that shape growth rates within our system (lines 408-444).

4. I think the authors lack a negative control where there is no spatial structure (i.e., there exists only one well). If the environment is fixed as (1,1,1,1), I guess the sky blue strain (HIS) dominates and this strain would not go extinct. However, if the environment fluctuates with some probability as in Fig.4 by the feedback as in Fig.5, how long does the community survive (I mean, when all of the four species go extinct)? Such control simulations would deepen the understanding of the results of Figs. 4 -6 since the spatial structure affects the species coexistence and extinction.

This is an excellent suggestion. Our current work stems from our previous experimental work where we manipulated the environments in a single well (Denton & Gokhale, 2020, mSystems). We show now in the supplementary material the results of having a single well instead of a structured ecology. The impact of not having a structure is then abundantly clear. All the lineages go extinct irrespective of the start conditions of the time difference between the pool and the well phase. We hope the inclusion of this detail in the SI acts as a good control case along with our previous work. The implementation of this null model is now also available on GitHub and for reproducibility.

Minor comments

1. It seems that the colors in Fig.1 does not match the explanations in the legend. For example, the authors wrote the ecology (or the environment) of the top left well (I believe {1,1}) is (1,0,0,0). According to the color bar on the top of Fig. 1, this should correspond to blue, but the color of the circle on the left column of Fig. 1 is purplish (I believe this environment is (1,1,0,0)). The colors at {1,4} do not match the legend either. I believe I am not color-blind. Please check the figure colors and the legend.

We apologise for this mistake (we're unsure how it happened). The text has been updated to reflect the actual ecology/community in cells {1,1} and {1,6}.

2. I would like the authors to clarify the definition of the strains (l. 127-128, l. 138-139, Fig. 1). Do the colors in the right column of Fig. 1 correspond to which metabolites are produced by each strain or community? Although the main text defines the strains by which metabolite each strain produces, Fig.1 does not explicitly explain what the colors of the communities (strains) mean.

We have updated the text and the figure legend to make this clear.

3. The font sizes in the pie charts in Figs. 5 and 6 are too small to read. I am young (in academia), but it is very hard to read the letters there.

We apologise for incorporating an optometry exam in our manuscript. The font sizes have been increased.

4. I do not think what is written in l. 256-258 matches the top panel of Fig. 6. Fig. 6 shows the effect of the duration of the pool phase, while the text explains the effect of the timing of the feedback.

This section has now been rewritten and figure references fixed.

Typos

l. 122 "Nature" => "nature" (I do not think the authors are talking about the specific journal)

True. Updated

l. 123 "analysis, are" => "analysis are"

Updated.

The left-hand-side of Eq (1) should be \dot{x}_i . Also, this equation lacks the death rate d .
The left-hand-side of Eq (2) should be \dot{x}_i .

We used the `\dot{}` in the latex template but it did not render. We have switched to Leibniz notation.

December 16, 2022

Dr. Jai A Denton
Monash University
World Mosquito Program
12 Innovation Way
Clayton 3800
Australia

Re: mSystems00929-22R1 (Ecological Drivers of Community Cohesion)

Dear Dr. Jai A Denton:

Your manuscript has been accepted, and I am forwarding it to the ASM Journals Department for publication. For your reference, ASM Journals' address is given below. Before it can be scheduled for publication, your manuscript will be checked by the mSystems production staff to make sure that all elements meet the technical requirements for publication. They will contact you if anything needs to be revised before copyediting and production can begin. Otherwise, you will be notified when your proofs are ready to be viewed. During this process, please make sure to include a separate paragraph on "Data Availability" and aggregate relevant codes and data links there for improved accessibility. It is also a good general practice to include a section on Author Contributions.

Publication Fees:

If you would like to submit a potential Featured Image, please email a file and a short legend to mSystems@asmusa.org. Please note that we can only consider images that (i) the authors created or own and (ii) have not been previously published. By submitting, you agree that the image can be used under the same terms as the published article. File requirements: square dimensions (4" x 4"), 300 dpi resolution, RGB colorspace, TIF file format.

We recognize that the video files can become quite large, and so to avoid quality loss ASM suggests sending the video file via <https://www.wetransfer.com/>. When you have a final version of the video and the still ready to share, please send it to mSystems staff at mSystems@asmusa.org.

Sincerely,

Babak Momeni
Editor, mSystems

Journals Department
Supplemental Text 1: Accept

Fig SI.7: Accept

Fig SI.1: Accept

Fig SI.3: Accept

Fig SI.4: Accept

Fig SI.2: Accept

Fig SI.5: Accept

Fig SI.6: Accept